robotics/biomimetics

self-assembly, self-folding, meta-material, magnetic interaction, magnetic catalysis, bottom-up manufacturing

**Author for correspondence:**
Shuhei Miyashita
e-mail: shuhei.miyashita@sheffield.ac.uk

# Catalytic self-folding of 2D structures through cascading magnet reactions

Emily J. Southern[1,2], Valentin Besnard[3], Bastien Lahaye[3], Andy M. Tyrrell[1] and Shuhei Miyashita[1,4]

[1]Department of Electronic Engineering, University of York, Heslington, York YO10 5DD, UK
[2]Department of Mathematics, Imperial College London, London, UK
[3]Department of Software and Systems, ESEO, Angers, France
[4]Department of Automatic Control and Systems Engineering, University of Sheffield, Western Bank, Sheffield S10 2TN, UK

 EJS, 0000-0002-2647-1885; SM, 0000-0002-9795-9247

While thousands of proteins involved in development of the human body are capable of self-assembling in a distributed manner from merely 20 types of amino acid, macroscopic products that can be assembled spontaneously from 'alive' components remains an aspiration in engineering. To attain such a mechanism, a major challenge lies in understanding which attributes from the bio-molecular realm must be leveraged at the macro-scale. Inspired by protein folding, we present a centimetre-size 1D tile chain whose self-folding processes are directed by structure-embedded magnetic interactions, which can theoretically self-assemble into convex 2D structures of any size or shape without the aid of a global 'controller'. Each tile holds two magnets contained in paths designed to control their interactions. Once initiated by a magnetic unit (termed Catalyst), the chain self-reconfigures by consuming magnetic potential energy stored between magnet pairs, until the final 2D structure is reached at an energetic minimum. Both simulation and experimental results are presented to illustrate the method's efficacy on chains of arbitrary length. Results demonstrate the promise of a physically implemented, bottom-up, and scalable self-assembly method for novel 2D structure manufacturing, bridging the bio-molecular and mechanical realms.

## 1. Introduction

Researchers have sought for centuries to understand how living creatures are capable of growth, reproduction and repair. Most current engineering techniques make use of a 'top-down',

pick-and-place style assembly process, where intelligence for how components assemble is held primarily by the assembling agent rather than by the components themselves. By contrast, inside the human body, thousands of different protein types can be produced from only 20 types of amino acid in a massively parallel synthetic process. They are able to dynamically develop and sustain their structures through layers of regulation encoded within each molecule and cell, acting simultaneously as both assembly agent and component. Techniques used in biological self-assembly have not yet been developed for artificial manufacturing, due in part to the difficulty in translating the required naturally occurring attributes at the macro-scale. We believe that if these techniques were transferred successfully, then other attributes associated with biological systems, such as self-repair, could also be transferrable.

We now examine four main approaches in current self-assembly research to determine attributes that, from an engineering standpoint, we believe will be essential for emulating biological self-assembly in an artificial system.

## 1.1. Biomolecular self-assembly

In nature, attributes required for self-assembly have been refined over millennia through the process of evolution. One example of biological self-assembly can be found in the assembly of DNA nucleotide bases into the correct A-T and C-G pairs in a thermally dynamic process involving massive sampling of arrangements, demonstrating a high level of addressability. By using DNA directly as a component material, it is possible to take advantage of this innate addressable property during self-assembly. The ability for DNA strands to be selectively cut and glued together in order to form two-dimensional [1] and three-dimensional [2] nano-shapes has been robustly proven [3]. Similar works have demonstrated a variety of other DNA self-assembled structures, including functional devices such as motors [4]. The self-assembly output using this technique shows resilience to perturbations in the environment, and efficiency in the massively parallel process employed to build each structure. However, the technique is limited in terms of assembly materials and the ability to scale upwards in terms of structure size, thereby restricting the technique to niche use cases.

## 1.2. Mechatronic self-assembly

Conversely, mechatronic self-assembling systems suffer from difficulties in scaling downwards due to the large volume of electronic parts that each module typically contains. A stochastic method of assembly coupled with a turbulent environment is typically used to initiate self-assembly [5]; the low frequency of connection attempts between components increases the time needed for assembly, and therefore reduces system efficiency. The modules' connections are controlled by either on-board micro-controllers or an external computer [6] to carry out decision-making on how modules should assemble. Mechatronic systems are highly programmable, which allows them to produce a large variety of final structures from a set of physically identical modules [7–10]; previous work has directly demonstrated practical capabilities such as LED displays and circuits [11,12].

## 1.3. Self-folding with smart materials

One method for handling the property of addressability in a system with many degrees of freedom is by initiating self-assembly from parts in an already connected state and then letting it form a three-dimensional structure [13]. Some of the models based on this approach are influenced by origami techniques whereby a sheet made of a smart material self-folds into its pre-programmed final state, thus allowing a large variety of final products [14–16]. Materials are 'programmed' with methods such as shear stress by using pre-stretched [17–19] or thermally responsive [20,21] materials, or pneumatic actuation [22]. However, actuation can typically occur once and for limited configurations only, reducing the variety of structures that one design can produce. An alternative method to overcome this lack of programmability is to include electronic components within the smart material, at the cost of scalability [13,14].

## 1.4. Mechanical self-assembly

Mechanical self-assembly techniques typically rely on stochastic interactions between modules to provide both programmability and addressability. Some works using a mechanical approach have exploited the

external geometry [23] and material composition [24,25] of modules, along with mechanically realized internal states [26], to influence components' interactions and addressability. Approaches focusing on internal mechanisms can cause components to only be addressable once a component has been activated in some way by another, allowing for the assembly sequence to be physically encoded within the component itself [27–29]. Patterning the surface of three-dimensional modules [30] or coating different faces with hydrophobic and hydrophilic materials [31,32] can also be used to increase the success rate for module addressability and produce homogeneous patterns in the final structure. Due to advances in micro-fabrication technology such as photolithography, mechanically grounded self-assembly can be scaled down to the micro-scale by using properties of the system's environment such as remote magnetic fields [33–36], electrostatics [37], microfluidic channels [38] and mechanical vibrations [39] to externally provide the components with stochasticity and kinetic energy. However, these systems rely on the probabilistic outcome of component interactions to achieve rarer configurations, and as such are subject to failure states if stochasticity in the environment is reduced.

Based on our analysis of the positive and negative aspects of current artificial self-assembling techniques, we define four attributes that we believe are essential features for emulating the traits of biological self-assembly in an artificial system: (i) material choosability, types of materials deployable by the system; (ii) material programmability, how seemingly identical components can exhibit different behaviours; (iii) material addressability, how a system determines which components should and should not be located next to each other within the final structure, in order for it to produce the desired configuration; and (iv) method scalability; the ability to decrease component size while maintaining the validity of the design principle. We also aim for deterministic design over a stochastic environment to increase the overall success rate.

## 1.5. Anfinsen's thermodynamic hypothesis

In contrast to the four artificial self-assembly approaches listed above, biomolecular self-assembling systems encounter fewer failure states by relying on a combination of both thermally dynamic environments and component features to determine how self-assembly will progress. A fundamental example of biological self-assembly is a protein's ability to reliably fold into a final 'native' state that is globally energetically stable. Anfinsen [40] was the first to hypothesize that all of the information required for a protein amino acid sequence to fold is contained with the chain itself and that the globally stable (native) state of a sequence must be located at the sequence's energetic minimum. As the number of possible folding configurations for a protein increases exponentially [41], it is clear that many possible configurations are automatically discarded as the protein folds into its native state. Anfinsen hypothesized that the following system requirements are necessary for a protein to reach its native state successfully:

(i)   Kinetical accessibility: The sequence must not require an additional source of energy on its route from its initial configuration in order to reach its globally stable state.
(ii)  Uniqueness: The sequence's globally stable state must not have any other configuration with a similar or lower level of free energy in its neighbourhood.
(iii) Stability: Minor perturbations of the system at its native state should not cause the system to reconfigure at a different energetic minimum. This can be pictured as a funnel-shaped energetic profile with the native state at the end of its spout, rather than a soup plate-shaped terrain with several states at a similar energetic level [42]. In this way, only a significant energetic input would be able to dislodge the system from its final position.

While Anfinsen's hypothesis is now widely considered to be a simplification of biological processes and that several counter-examples to his proposed system requirements exist, we believe that his statements are a useful starting point for our approach in micro-robotics. Based on Anfinsen's hypothesis, this work addresses our hypothesis that self-assembly at higher-order degrees of freedom, as illustrated in figure 1, can be handled without stochastic characteristics but through mechanically attained chain reactions. We are interested in developing a system where shape reconfiguration can be carried out by forming a funnel-shaped energetic profile, and then initiating and transmitting it through a catalytic process. There exist several attempts to restore a three-dimensional (3D) shape from a stretched one-dimensional (1D) chain [43,44] but, as the self-folding at joints can only occur simultaneously, chain self-collision limits the number of configurable shapes. To remove the chance of collision during self-assembly, we use a process of uniaxial folding where the chain folds in alternating directions along a

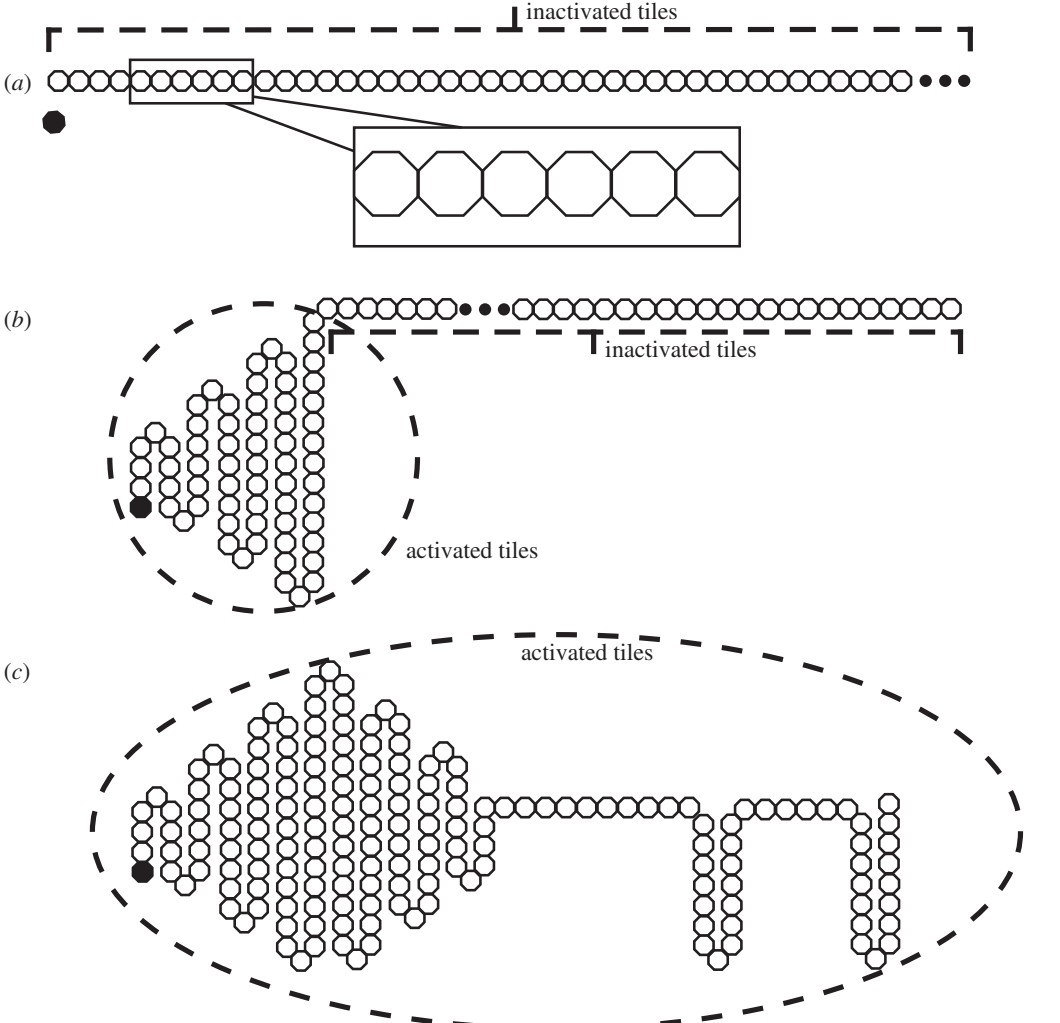

**Figure 1.** The proposed self-folding approach showing the stages of chain folding from 1D to 2D. (*a*) Section of the chain in its initial state. All tiles have an octagonal shape. (*b*) Chain during folding process. Folds occur sequentially once the chain has been triggered by Catalyst (in black). (*c*) The completed shape (a key).

folding axis, which also guarantees that the system can self-fold into any type of two-dimensional (2D) convex polygon.

## 2. Methods

### 2.1. Magnetic interactions

In our previous work, we used a combination of permanent magnets and the geometry of centimetre-sized components to demonstrate magnetically achieved enzymatic catalysis and inhibition, behaviours typically seen in biomolecular reactions [45]. The goal of this study is to extend this mechanism to develop a method of self-directed and decentralized manufacturing in which a one-dimensional chain spontaneously reconfigures into a two-dimensional structure through cascading interactions between magnets, as seen in figure 1. An outlook of the tile chain is shown in figure 2. The system is composed of a chain of buoyant octagonal tiles and a cylindrical component, Catalyst, which triggers magnetic movement. Both tiles and Catalyst can move freely on a horizontal planar surface (the surface of water). The tiles have two detachable hinges on two of their edges. Two paths inside each tile hold a pair of cylindrical permanent magnets in position. Magnets are placed with alternating north and south poles facing upwards throughout the system, and can slide horizontally along their paths while maintaining their posture, since each magnet is attracted to

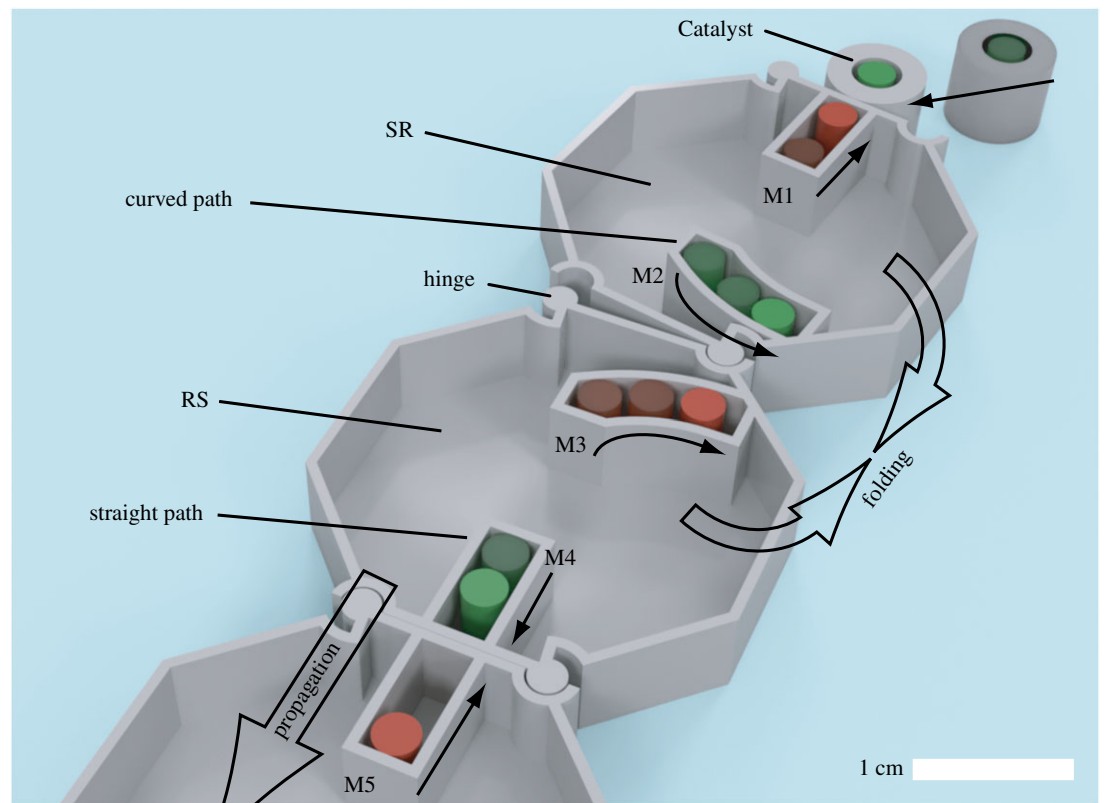

**Figure 2.** Composite of Catalyst's approach and initiation of the tile chain. Red and green colours of the magnets indicate north and south poles facing upwards; locations of north-facing and south-facing magnets does not matter as long as they are placed in alternating order.

both of its adjacent magnets in the chain. Tiles are considered 'inactivated' while the magnets they contain remain in their initial positions, and considered 'activated' once their magnets have begun to move.

The magnetic potential energy $U_{ij}$ produced by a pair of magnets $M_i$ and $M_j$ ($i \neq j \in \mathbb{N}$), with respective magnetic moments $m_i$ and $m_j$ (A m$^2$), in a vacuum is given by

$$U_{ij} = -\sigma_{ij} \frac{\mu_0}{4\pi} \frac{m_i m_j}{r_{ij}^3}, \tag{2.1}$$

where $\sigma_{ij} = 1$ if magnets have opposite orientations and thus are attracted to each other, and $-1$ if magnets have the same orientation and thus repel each other. $\mu_0 = 4\pi \times 10^{-7}$ (H m$^{-1}$) is the permeability of free space, and $r_{ij}$ (m) is the distance separating both magnets. The resultant magnetic force $F_{ij}$ (N) is therefore

$$F_{ij} = -\frac{dU_{ij}}{dr} = -\sigma_{ij} \frac{3\mu_0}{4\pi} \frac{m_i m_j}{r_{ij}^4}. \tag{2.2}$$

For $N$ magnets ($i, j \in N$), the total potential energy of the system, $U_{\text{total}}$ (J) is defined as

$$U_{\text{total}} := \sum_{i<j} U_{ij}. \tag{2.3}$$

Due to the rapid decline of magnetic force over space, we only consider adjacent magnet sets for the system design and we define $U'_{\text{total}}$ (J) as the corresponding potential energy.

System behaviour is dependent on the type of path shape chosen for each tile. Paths in which magnets are located are chosen to be either straight, curved to the right, or curved to the left, as viewed from above (figure 2). The curved paths are designed with a short straight section, followed by a circle arc such that the relative distance between adjacent curved paths continuously decreases. Two attracting magnets can transit to the direction where the gradient of magnetic potential energy is negative. If the paths located at either side of a hinge are straight, then neighbouring magnets in consecutive tiles will attract and

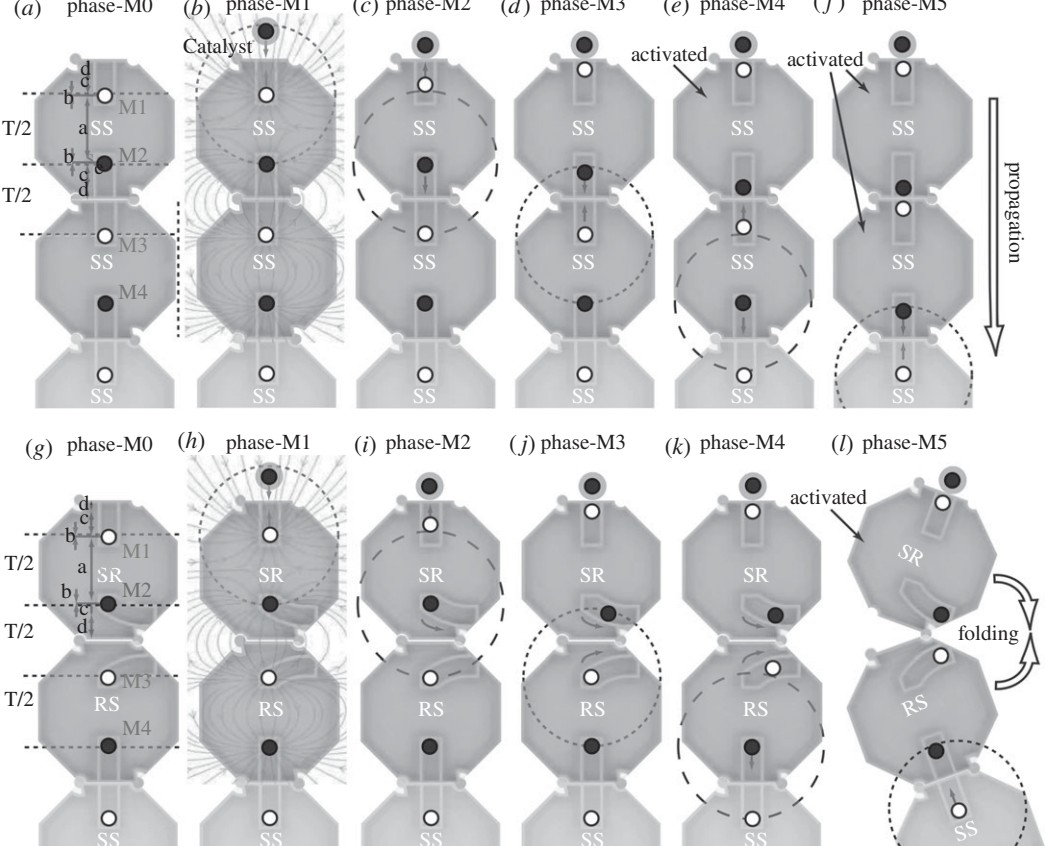

**Figure 3.** Stages of system initiation and propagation for (*a–f*) straight paths and (*g–l*) curved paths. (*b,h*) The magnetic flux lines caused by the chain magnets to demonstrate how Catalyst can approach the chain. Circles indicate the relative distances between magnets at different phases: dashed circles have diameter $T/2 + 2b$, dotted circles have diameter $T/2 - 2b$. Arrows show which magnets are currently in motion.

decrease the distance between themselves by transitioning their positions directly towards each other. Alternatively, magnets travelling along a pair of curved paths will produce torque about the hinge and cause the tiles to change contact surfaces (both will be explained in detail in §2.2). We call such interactions between magnets 'magnetic reactions' hereafter.

## 2.2. Path design

Figure 3 shows the phases of magnetic reactions for two different tile configurations: figure 3*a–f* depicts phases during a linear propagation of magnetic reactions between tiles, while figure 3*g–l* depicts a propagation of magnetic reactions that will induce folding of the chain. Each phase signals the point in time at which a magnet will begin to move: Phase-M1 corresponds to when M1 begins moving, Phase-M2 corresponds to when M2 begins moving, etc. Figure 3*a,g* shows the system's initial states. In figure 3*b,h*, Catalyst is released in the region where it is attracted to M1 and activates Tile 1. Magnetic flux lines show trajectories which Catalyst follows. Dashed circles depict points in time where a magnet has become more attracted to one neighbouring magnet than the other due to the movement of one of its neighbours. After the initial movement of both Catalyst and M1, M2 will begin to move towards M3 as it is now its closer neighbour, shown in figure 3*c*. M2 then triggers the movement of M3, and thus activates Tile 2 (figure 3*d*). Once M3 has moved towards M2, M4 then begins to move, triggering the movement of M5 and the activation of Tile 3, as shown in figure 3*e,f*. This cascade-type process will continue until the end of the chain is reached. Note that a similar magnetic interaction was proposed in [29], in which magnets are placed horizontally and no curved path types or self-folding was proposed.

Figure 3*h–l* demonstrates the same process as in figure 3*b–f*, but with M2 and M3 held inside curved paths that allow folding at the hinges between their respective tiles. While viewing figure 3*j*, it can be seen that M2 is closest to the initial position of M3 when it is partway down the length of its path; thus, the curved path must be designed such that M3 will begin moving either before or when M2

reaches this point in order for the chain reaction to continue. In figure 3*k*, M3 has been initiated and then moves to the mirror image point of M2 in its own path. M2 and M3 then move along their respective paths together as the distance between them continues to decrease. The folding torque shown in figure 3*l* is produced once magnets have passed the centre of one of the tile hinges. Sections of chain before and after the hinge rotate around the hinge point as a unit. The tile geometry causes rotation to be halted once a 90° angle between the pre-fold and post-fold chain sections has occurred. The reaction is completed when all magnets in the chain have reached their final positions.

In contrast to weak attractive forces between inactivated tiles in their initial chain configuration, the final configuration of activated tiles is held in place by strong magnetic forces between pairs of magnets in close proximity to each other. Therefore, the chain's final self-folded configuration is fixed. In addition, if tiles are manually separated after reaching their final configuration, magnets automatically return to their initial positions due to the inter-tile magnetic attractions and the path geometry allowing their return. Consequently, the tiles can be instantly re-used as part of a different chain configuration.

Relying on tile symmetry, only four tile types are necessary for the entire system:

— straight path–straight path (SS)
— straight path–left turn path (SL)
— left turn path–straight path (LS)
— left turn path–left turn path (LL).

We find right turn path–straight path (RS), straight path–right turn path (SR) or right turn path–right turn path (RR) by rotating tiles (SL), (LS) and (LL) 180° in the x–y plane, respectively. LR (equivalently RL) is also a possible tile type; however, as it produces diagonal lines which are not necessary in the current system design, it is not used here. Tiles are placed in the chain such that paths on each side of a hinge are complementary—SS/SL/LS is a possible chain configuration, while SS/LS/LS is not. As each connection between tiles will produce one of three different outcomes, a chain consisting of $K$ tiles can produce up to $3^{K-1}$ different configurations before taking issues such as chain self-collision into account.

## 2.3. Geometric conditions for path design and coordination of folding timings

The geometric conditions on lengths and placement of magnet paths that ensure successful magnet initiation and reactions are derived here. As can be seen in figure 3*a,g*, $T\,(\geq 0 \in \mathbb{R})$ is the tile length and $a\,(\geq 0 \in \mathbb{R})$ is the distance between magnet initial positions in a tile. The distance from magnet initial position to the tile edge can be split into three sections: $b\,(\geq 0 \in \mathbb{R})$, which is the distance from the initial position to the line upon which magnets would be equidistant; $c\,(\geq 0 \in \mathbb{R})$, which is the distance from the equidistant line to the point on the path at which the next magnet in the chain would begin to move; and $d\,(\geq 0 \in \mathbb{R})$, which is the distance from the end of $c$ to the tile edge. $b$ is the same for both path types while values of $c$ and $d$ vary.

For magnets to remain in their initial positions before the system is initiated, we require

$$a \leq 2(b + c + d). \tag{2.4}$$

In order for the magnet in the following tile to begin its trajectory before the magnet in the previous tile has reached the end of its path, it is required that

$$b + c + 2d < a. \tag{2.5}$$

(2.4) and (2.5) can be combined to form

$$b + c + 2d < a \leq 2(b + c + d), \tag{2.6}$$

which determines required path length but not the required path location within the tile.

As the length of the whole tile $T$ is $a + 2(b + c + d)$, it can be seen from equation (2.4) that $a$ must be less than or equal to half of the total tile length, $a \leq T/2$. Initially, the intra-tile distance is $T/2 - 2b$ while the inter-tile distance is $T/2 + 2b$. A magnet must therefore move a distance of at least $2b + 2b = 4b$ before it will be capable of initiating the movement of the next magnet in the chain. The magnet path must be sufficiently long to allow this pre-initiating movement, and thus

$$4b < b + c + d. \tag{2.7}$$

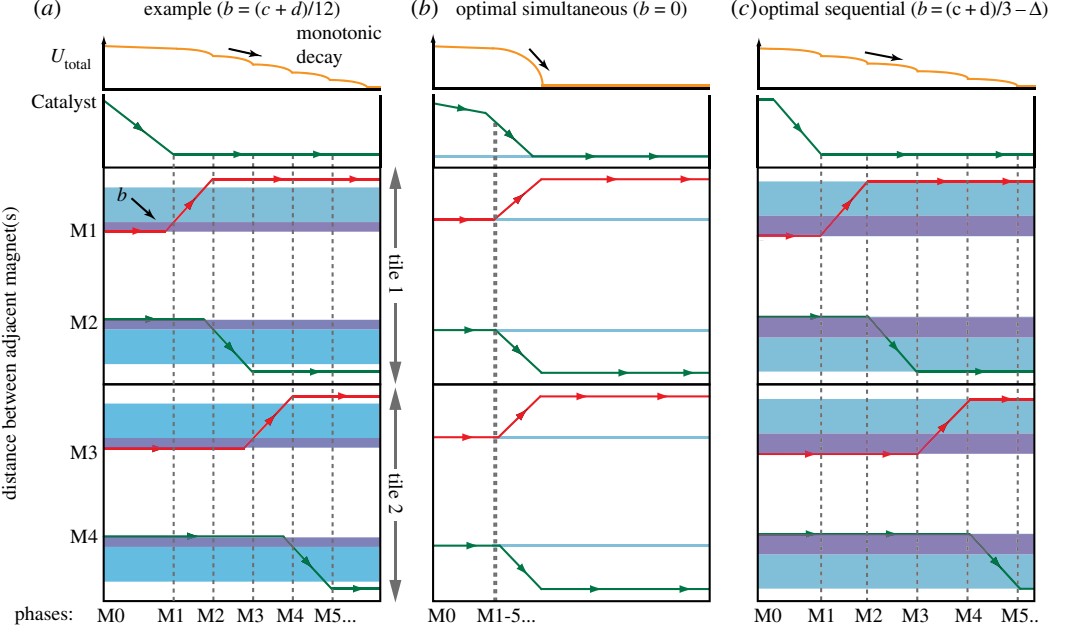

**Figure 4.** Magnet transition graphs representing the movement of each magnet once the system has been initiated. Vertical axes show distances between adjacent magnets; horizontal axes show the reaction timings (not necessarily evolving at a constant speed). Purple regions indicate length of $b$. Blue regions indicate the necessary region that a magnet must traverse before the next magnet can begin to move. Vertical dashed lines indicate when a magnet has reached its final position. Orange lines represent the potential energy for each type of system. (a) Example value of $b$ (in this case $b = (c + d)/12$) and accompanying traversable region and potential energy decay. (b) $b = 0$: the magnets are equidistant and need to travel only a minimal distance for the next magnet to be initiated. (c) $b = (c + d)/3 - \Delta$, where $\Delta$ is an arbitrarily small distance: the magnets must travel across almost the full length of the path before initiation of the next magnet is possible.

We finally obtain

$$0 \leq b < \frac{c + d}{3}. \tag{2.8}$$

Equations (2.6)–(2.8) provide geometric conditions allowing catalytic-type reactions between magnets.

By varying the length of $b$ within its allowable interval, triggering of the magnet motion can be regulated from quasi-sequential to quasi-simultaneous. Figure 4 demonstrates the system behaviour for three different values of $b$. Note that the actual magnet velocity is not constant, while here it is represented by straight arrows. Horizontal axes represent the timings of different phases, while vertical axes represent distances between adjacent magnets. Changes in the length of $b$ (represented with a purple band) alter the distance each magnet must travel (represented with a blue band) before the next magnet will begin to move:

— A value of $b$ somewhere within its range of possible values will produce a system where magnetic propagation overlaps to a degree dependent on the chosen size of $b$ (figure 4a).
— When $b$ is taken as its minimum possible value, 0 mm, magnets will in theory be actuated as soon as the previous magnet in the chain begins to move, thus producing a quasi-simultaneous self-assembling system (figure 4b).
— If $b$ is taken as its largest possible value, slightly smaller than $(c + d)/3$, then a system where magnets are actuated quasi-sequentially is produced (figure 4c).

## 2.4. Magnetic catalysis

Before the system reaction has been triggered by Catalyst, it can be considered at a local energetic minimum, since all magnets are as close as physically attainable to the magnet they are being pulled towards. Once Catalyst triggers a reaction, the system consumes the magnetic potential energy and reaches a new energetic minimum—where magnets are at final positions at the end of their respective

paths. A representation of the potential energy decrease for different lengths of $b$ is shown above transition graphs in figure 4a–c. Without Catalyst, such reactions will not occur unless M1 moves against the decay incentive of potential energy. Paths for magnets in this system are designed to act as both a physical and energetic pathway for the magnet to travel along, such that magnets can reach the lowest available energetic minimum equivalent to the final configuration state through a process of cascadic catalysis. The system presented here thus fulfils all conditions stipulated by the thermodynamic hypothesis. We would like to emphasize that, when the conditions are fulfilled and magnets proceed with reactions, a monotonic decay of the system's potential energy is guaranteed. As monotonic decay does not occur when tiles exist separately, there is a magnetically created 'activation' (threshold) potential found when forcing the magnets to move in other directions (i.e. increasing the distance between attracting magnets). We regard this planarization of potential energy as 'magnetically achieved catalysis'.

# 3. Results

This section first shows results from experiments using 3D-printed tiles, followed by simulation results for hundreds of tiles.

## 3.1. Experiments

Experiments were conducted with up to 10 tiles (30 mm length × 30 mm width × 7.5 mm height) and Catalyst (9.8 mm diameter × 7.5 mm height) placed inside a 40 cm diameter water tank. Both tiles and Catalyst were designed using Autodesk Fusion 360 and 3D-printed using a Stratasys Connex 3 Systems Object500. The tank was filled to 5 mm height with water in order to provide buoyancy and minimize friction between the tile base and tank. 5 mm height, 3 mm diameter cylindrical neodymium magnets (Supermagnete), with magnetic moment $m = 0.4445$ A m$^2$ are placed within the paths of each tile. Tile paths were filled with water and magnets were dipped in mineral oil (Johnson & Johnson) prior to being inserted into the tile paths, in order to reduce friction between the magnet sides and path walls. Application of oil and water allows the system to approximate an ideal mass-less (through buoyancy), friction-less (through lubrication) system, where the existence of a minimal amount of torque around a hinge will induce folding.

Figure 5 shows Experiments I and II, for two different chain configurations composed of three tiles (see electronic supplementary material, Video SI and Video SII for an experiment containing seven tiles): Experiment I (figure 5a–g) is a system consisting of linear propagation followed by a 90° folding, while in Experiment II (figure 5h–n) two consecutive 90° turns are used to form 180° folding. In theory, the maximum possible value of $b$ for the tile dimensions used in this work is $1.694/3 = 0.565$ mm. The value of $b$ used in this study was chosen under practical considerations to be 0.3 mm, equivalent to one tenth of each magnet's diameter. Although this value appears to be negligibly small, it produces a ratio between intra-tile and inter-tile magnetic forces of approximately 7 : 5 that allows the magnets to remain stably in their initial positions even while tiles are arranged into a chain.

In figure 5a, Catalyst is attracted to M1 when it is released by hand near to the first tile in the chain. Catalyst then initializes the movement of M1 (figure 5b), which is taken to be time $t = 0s$ in the system's catalytic process. Figure 5c shows M1 in its final position and M2 partway along its path. M2 assumes the role of Catalyst with respect to M3 and begins the movement of M3, shown in figure 5d. In figure 5e, M4 moves towards M5 within the curved path. In figure 5f, M4 has moved close enough to M5 to trigger the movement of M5. M4 and M5 continue to reduce the distance between each other by travelling along their respective paths, and produce torque around the hinge once they have travelled past the centre of the hinge. The last magnet in any chain will not move as there is no other magnet for it to move towards. The final, stable configurations are shown in figure 5g. An almost identical process is shown in figure 5h–n, except that M2 and M3 are also contained within curved paths. The phases occur within fractions of seconds of each other. All four possible configurations for three tiles were tested five times each, with 100% success rate for every configuration.

Figure 6 shows the system magnetic potential energy $U_{total}'$ from both experiments (calculated from adjacent magnets only) as the catalytic reaction is being carried out. Magnet positions were tracked using motion tracking software (Tracker). Magnetic potential energy is shown to be decreasing almost monotonically throughout reconfiguration with some areas of near stability. Variation is caused by the different magnet paths holding M2 and M3, for linear propagation and for propagation that causes

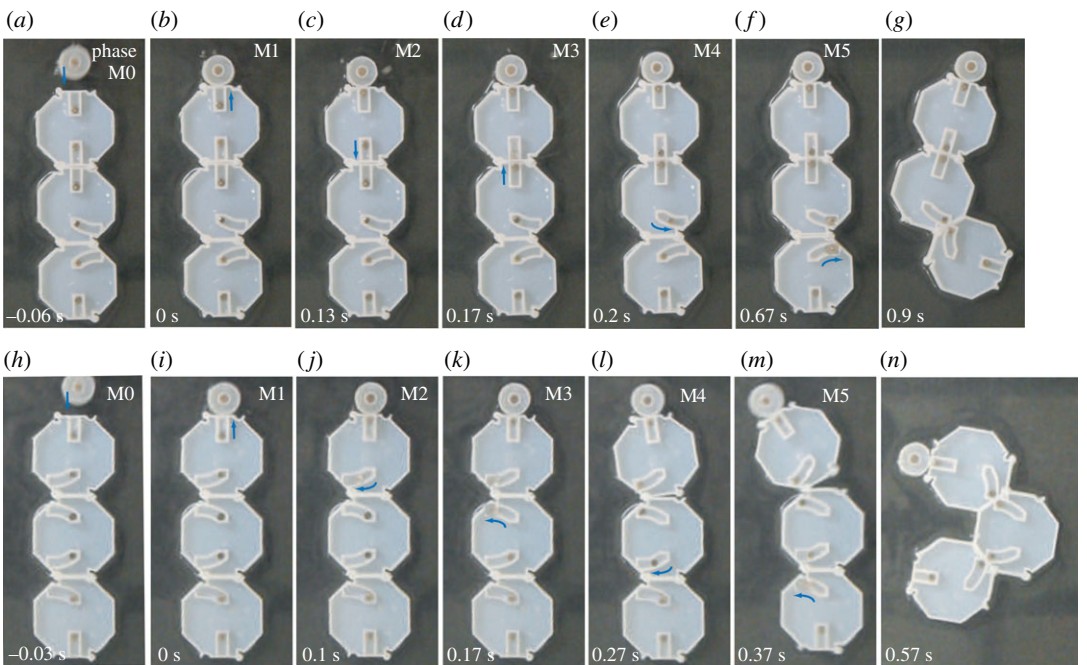

**Figure 5.** Experiments showing two different reactions with three tiles. Values in upper-right corners indicate the approximate phase in each image. (*a–g*) Experiment I: Linear propagation between tiles followed by a 90° turn. (*a*) Catalyst is attracted to and moves towards M1. (*b*) Once Catalyst makes contact with the first tile, it attracts M1 enough to begin M1's movement. (*c*) M2 is able to start moving towards M3 once M1 has travelled a distance equivalent to 4*s* along its path. (*d*) M2 assumes the role of Catalyst in beginning the movement of M3. (*e*) M4 begins moving along its curved path. (*f*) M4 assumes the role of Catalyst for M5 from partway along its path. (*g*) M4 and M5 trigger a folding motion between the second and third tiles. (*h–n*) Experiment II: Two consecutive 90° turns to produce a 180° turn. The process is very similar to (*a–g*) but M2 and M3 are held within curved paths. Note that two folding motions can occur with some overlap due to their proximity and the chosen length of *s*. See electronic supplementary material, Video SI for further detail.

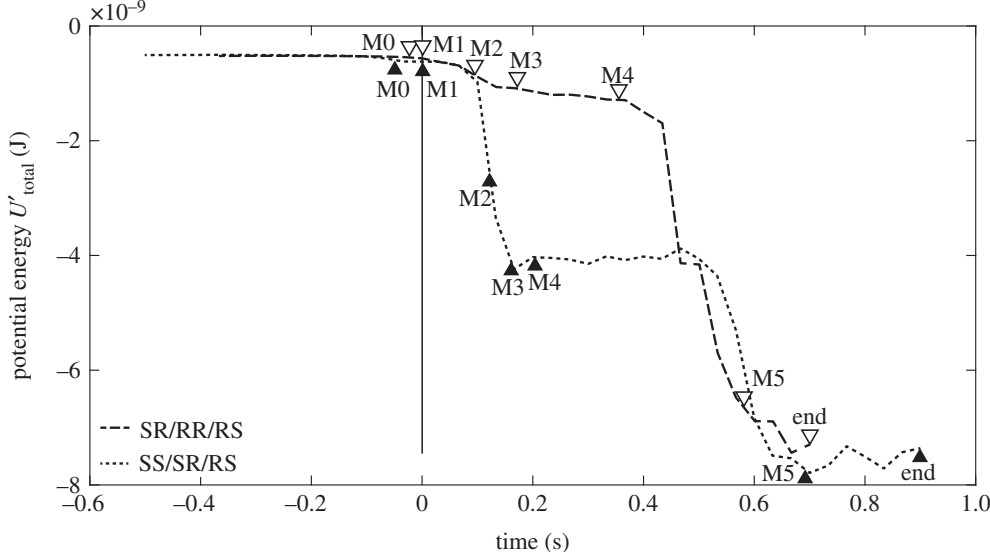

**Figure 6.** Comparison of magnetic potential energy decay (in Joules) for an SS/SR/RS configuration shown in figure 5*a–g* and an SR/RR/RS configuration shown in figure 5*h–n*. Solid vertical line indicates the point at which M1 begins moving (Phase 1). Black and white triangles align approximately with images shown in figure 5.

folding. Fluctuations after 0.6 s are caused by momentum acting upon Catalyst once the folding motion has been completed. As designed, the potential energy in both experiments is approximately the same before and after reconfiguration as the same number of tiles are used in each experiment.

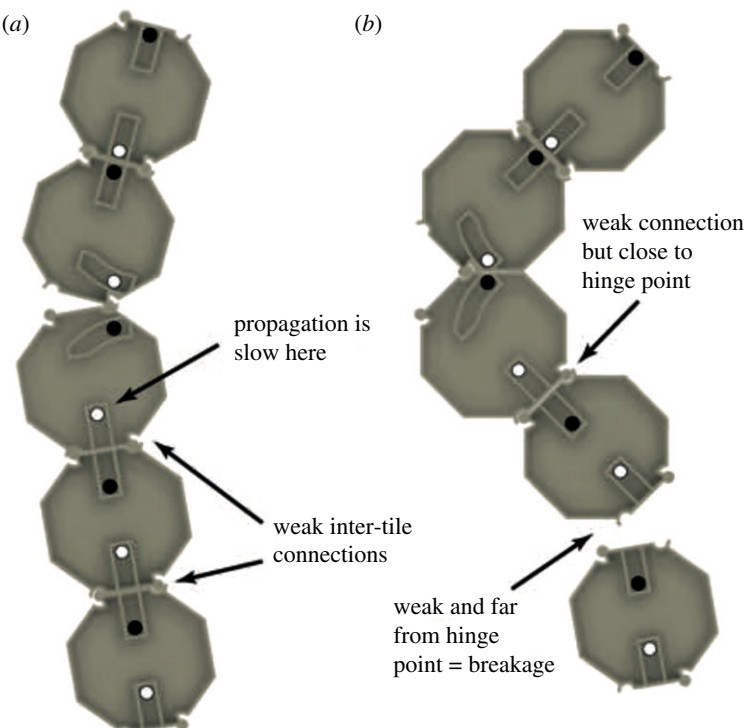

**Figure 7.** Example of chain breakage in a five-tile experiment. (*a*) Folding is occurring between the 2nd and 3rd tiles but magnet reactions are yet to reach tile 4. The connections between tiles 3 & 4, and tiles 4 & 5, are therefore weaker. (*b*) Folding is nearly completed between tiles 2 and 3 but propagation is still slow. The combination of a weak connection between tiles 4 & 5 and their physical distance from the hinge point causes breakage to occur, as tile 5 does not rotate as a unit with tile 4.

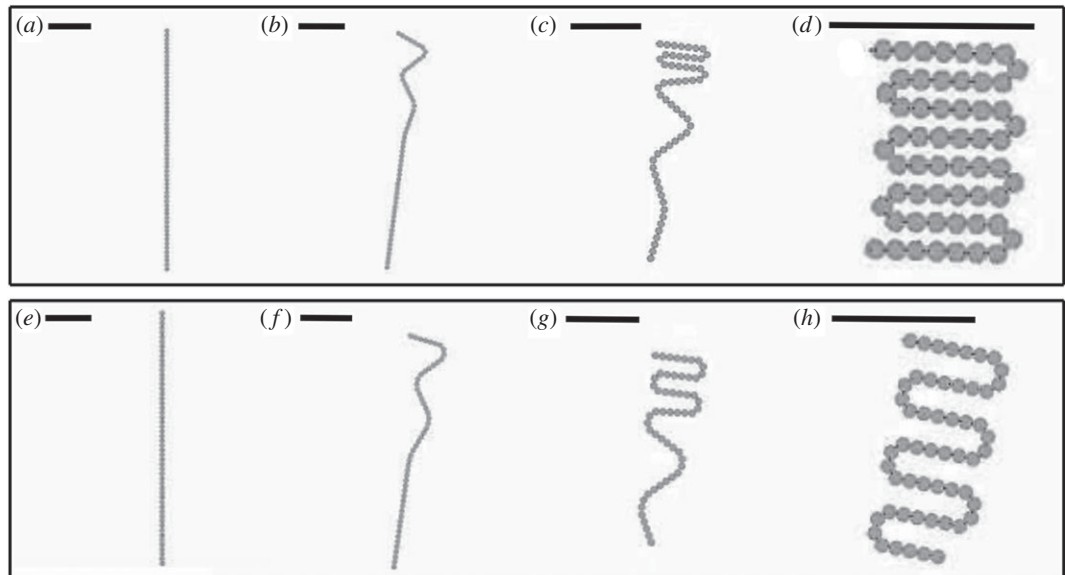

**Figure 8.** Two simulations of chain actuation and reconfiguration. (*a*–*d*) 57-tile rectangle composed of four tile types with higher tile density. (*e*–*h*) 54-tile rectangle composed of three tile types at a lower tile density. Scale bars represent the length of 10 tiles.

We observed failure cases where the chain split apart while carrying out experiments with larger numbers of tiles, as depicted in figure 7. This tended to occur if propagation occurred slowly relative to folding speed due to variables such as friction. During folding, the tile sections on each side of the hinge point experience an amount of torque and rotation that is dependent on the section length. The tiles in the section before the folding location are strongly connected together as they have been activated, but tiles situated after the hinge point may not yet have been activated if propagation has not reached them yet and will therefore be less strongly connected (figure 7a). During experiments,

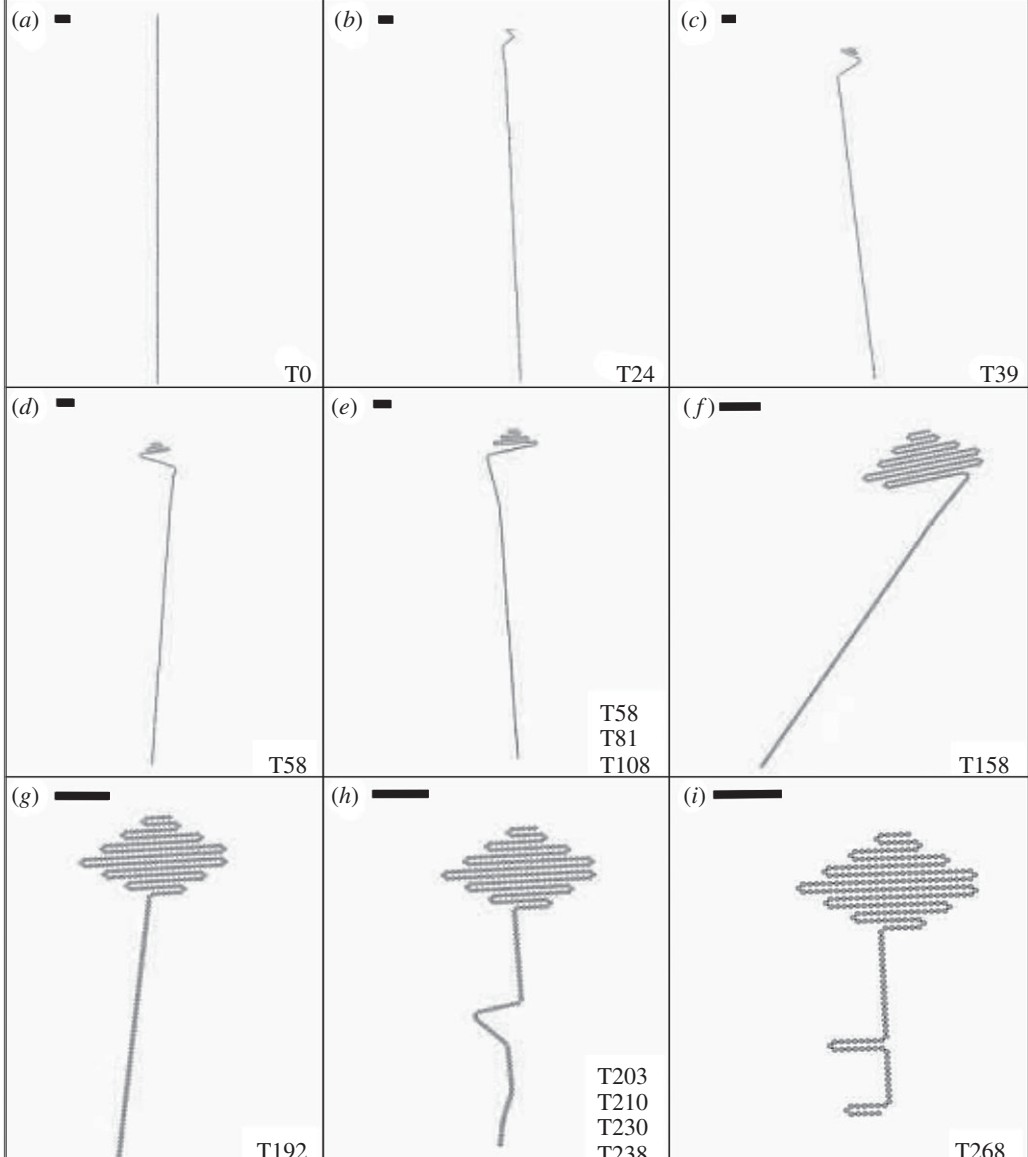

**Figure 9.** Frames from a simulation of 268 tiles (536 magnets) folding into the a key shape, taken from electronic supplementary material, Video SIII. (*a*) Initial configuration. (*b*–*i*) Stages of the folding process. Scale bars represent the length of 10 tiles. Values in bottom-right corners indicate the tiles where folding is taking place.

the chain could separate into two pieces at a distance two or three tiles away from the hinge point as the inactivated tiles attempted to rotate a significant amount as a unit during folding, as shown in figure 7*b*. The probability of failure decreases as the distance between the hinge point and the point which propagation has reached increases.

## 3.2. Simulation

Simulations were carried out using Blender Game Engine to investigate the scalability of the system. We also developed a MATLAB code which can take a 2D image and produce the tile sequence required to create the shape at a prescribed pixel resolution. Tiles are modelled as having neither mass nor friction against the ground as in an ideal system, while magnets are provided with mass so that its motion dynamics can be reproduced with greater accuracy.

Figure 8*a*–*d* shows the self-folding of a rectangle composed of 57 tiles and 115 magnets. Figure 8*e*–*h* shows the self-folding of another rectangle composed of 54 tiles and 109 magnets, but with lower tile density due to the change in tile sequence. Both simulations are initiated by Catalyst. In both simulations, folds occur uniaxially in order along the chain until all magnets are in their final positions and the final shape is

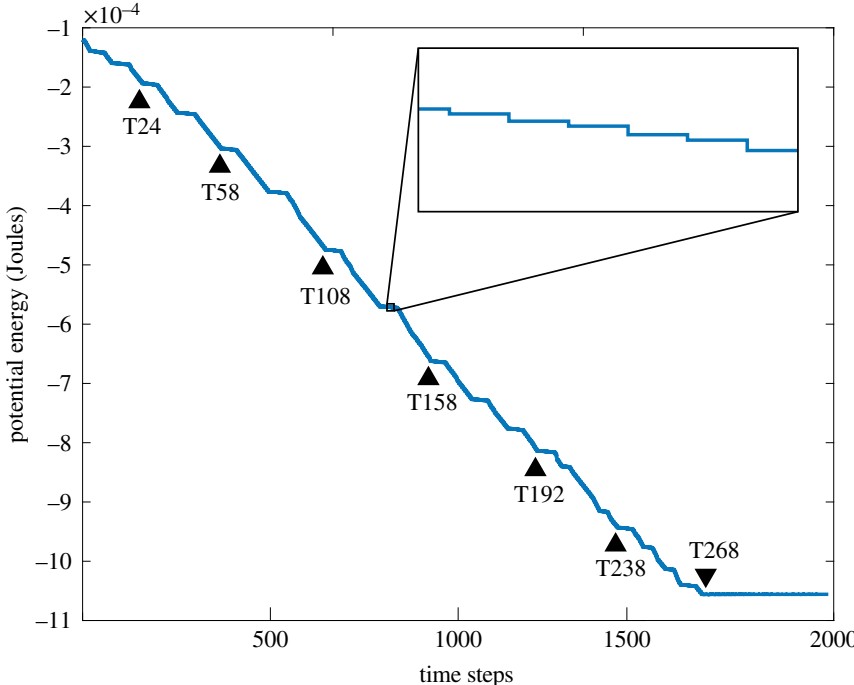

**Figure 10.** Magnetic potential energy decay calculated from the simulation in figure 9. Time steps marked with triangles align with images shown in figure 9a–i. Inset: zoomed-in portion of the plot shows decay occurring in a staircase-like fashion, similar to the potential energy decay calculated from experiments.

produced. The alternating clockwise and anticlockwise 180° turns of a uniaxially folding system ensure that issues of chain self-collision do not need to be taken into account during shape design. The folding sections of figure 8a–d are composed of three tiles in alternating SR/RR/RS and SL/LL/LS configurations, which produce the tightest possible turn for this tile design. Meanwhile, the folding sections of figure 8e–h alternate between SR/RS/SR/RS and SL/LS/SL/LS configurations, which produces a final structure with a lower tile density.

Figure 9 and electronic supplementary material, Video SIII depict self-folding of a key shape using 268 tiles, 536 magnets and Catalyst, demonstrating the system's ability to produce a functional geometry. The tile where folding is taking place is labelled at each stage. It can be seen that there are times when multiple folding motions are occurring simultaneously; this does not cause collisions due to the uniaxial nature of the folding process. The structure includes a single 90° turn instead of two consecutive turns to produce the shaft of the key.

Figure 10 shows the transition of potential energy $U_{total}'$ (in Joules) for the key simulation presented in figure 9. It can be seen that the potential energy decreases approximately linearly overall. The magnetic chain reaction involves a catalytic process, as shown by the flattened activation potentials and the 'step-like' shape of decay from the magnetic potential energy being inversely proportional to the cubic distance between two magnets. Steeper decreases in potential energy are located where multiple folding motions correlate significantly.

Figure 11 and electronic supplementary material, Videos SIV and SV show self-folding structures as proof of the system scaling linearly to a larger number of tiles: a saw constructed from 684 tiles (figure 11a), a glass shape constructed from 422 tiles (figure 11b) an arrow constructed from 302 tiles (figure 11c), and the numbers 1, 2 and 3 constructed from 191, 412 and 723 tiles, respectively (figure 11d–f). Cases where multiple folding actions were occurring in parallel, for example six simultaneous folds during the creation of the number 1, reduces the length of rotating chain and the consequent torque on inactivated tiles, thereby increasing system stability.

# 4. Discussion

## 4.1. Formational shapes

Theoretically, the method presented here of uniaxial folding is capable of producing all convex polygonal shapes. Concave polygons can also be produced, if they are monotone with respect to a folding axis: the

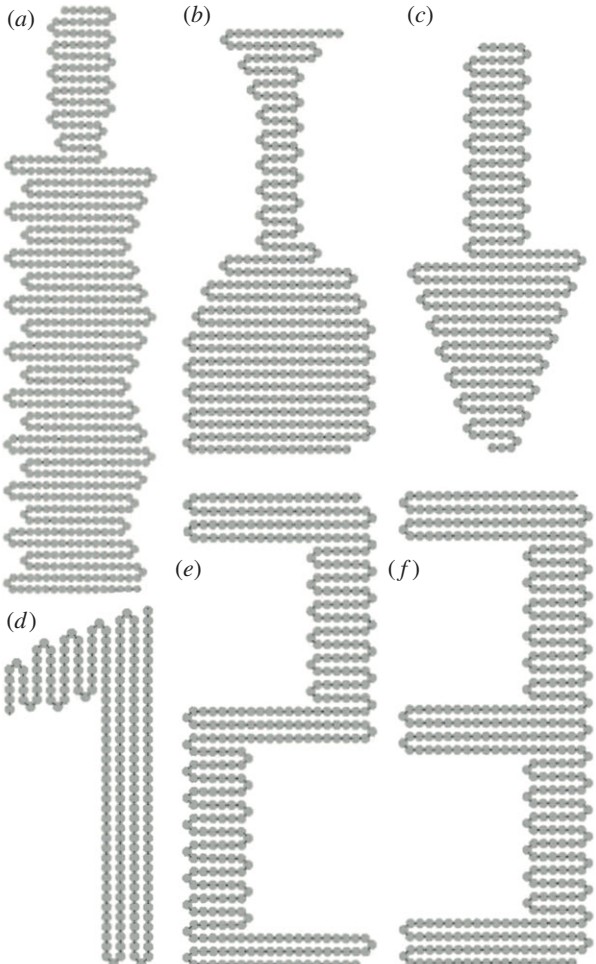

**Figure 11.** Larger shapes that can be self-folded using the same method with a linear increase in complexity. (*a–c*) Taken from electronic supplementary material, Video SIV. (*d–f*) Taken from electronic supplementary material, Video SV. (*a*) Saw shape constructed from 684 tiles. (*b*) Wine glass shape constructed from 422 tiles. (*c*) Arrow shape constructed from 302 tiles. (*d*) Number '1' constructed from 191 tiles. (*e*) Number '2' constructed from 412 tiles. (*f*) Number '3' constructed from 723 tiles.

crescent shape shown in figure 12 cannot be produced if folding occurs at the top and bottom of the shape with respect to a horizontal folding axis (figure 12*a*), but can be produced if they occur at the sides of the shape with respect to a vertical folding axis (figure 12*b,c*). The foldability for a non-monotone concave shape can be determined using an algorithm as described in [46].

## 4.2. Scalability

In theory, the number of tiles in a chain can be scaled up with a linear increase in complexity—for instance, an addition of one tile to the chain will increase the number of magnetic interactions by one also. In practice, as described in §3.1, if folding begins to occur while the interactions between magnets are still very close to the folding point, breakage could occur due to weak connections between inactivated tiles that must rotate a large amount during folding. All failure cases during experiments occurred due to the chain breaking apart. This could be fixed by making the hinge points more permanently connected; however, here we chose to keep the hinge design detachable in order to demonstrate the instantaneous reconfigurability of the system after actuation is completed. The magnet trajectories are geometrically determined and thus can be directly scaled down as the tile size is decreased. If the length of each tile wall is reduced by half, the force between magnets will decrease by a factor of 4 (equation (2.2)). Meanwhile, both tile mass and the moment of inertia at each hinge will decrease by a factor of 8. Thus, maintaining the relative position between magnets ensures that the forces produced by magnetic interactions will dominate over other forces in the environment, and thus the system holds at both larger and smaller dimensions. Coulomb friction is also proportional to tile mass until smaller scales are

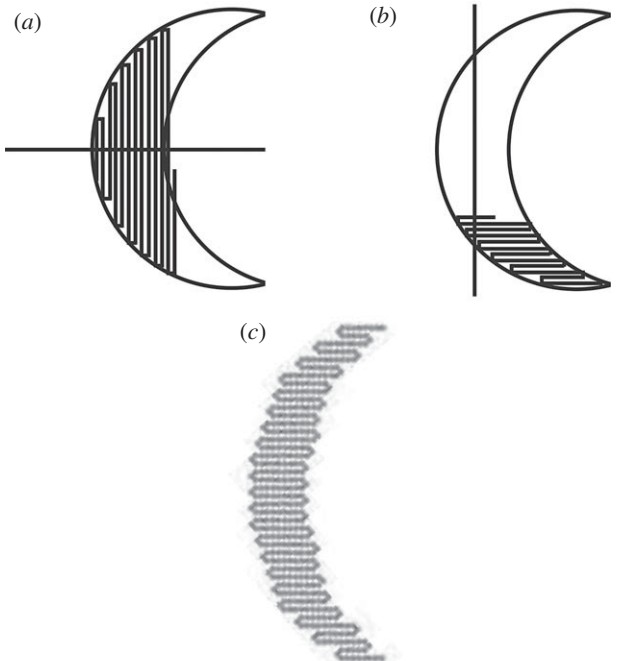

**Figure 12.** Examples of shapes that can be folded using the system: (*a*) The crescent shape cannot be folded using the horizontal line as a folding axis due to the concavity of the shape. (*b*) The crescent shape can be approximated using the vertical line as a folding axis. (*c*) Crescent shape constructed from 403 tiles.

reached, at which point further research will be necessary to understand the influence of friction (and sticktion) on magnet and tile dynamics as the scale is decreased further.

## 4.3. Potential applications

As long as the scalability aspects of the method hold, the system can be applied to smaller dimensions with different materials, making it suitable for manufacturing of micro-scale meta-materials and meta-structures. The system could be used directly to make a structure, or indirectly as a scaffold to reconfigure another material. Due to the amount of foldable configurations for a single chain, the system could spontaneously produce a wide variety of functional structures efficiently. Moreover, after each actuation, the tiles can be reset and made ready to form another chain by manually separating them, rendering it suitable for producing structures that are needed temporarily or would otherwise be disposed of after single usage.

## 5. Conclusion

This study presents a new approach for producing two-dimensional structures. The system uses a process of cascading magnetic catalysis to self-fold two-dimensional structures from a one-dimensional chain composed of tiles and magnets. We showed reliable and efficient self-assembly into a structure predetermined by the user, with the only human input being the addition of a reaction initiator (Catalyst) into the system environment. The chain of tiles self-folds uniaxially and the final structure grows such that a variety of two-dimensional shapes (all convex polygons, and concave monotonic polygons) can be produced in a bottom manner, from just four types of tile. Geometric conditions were derived that govern the required magnet pathways to ensure that catalysis along the chain is possible, and it was shown that the system can be tuned between quasi-simultaneous and quasi-sequential behaviour. The final output structure is designed to be at an energetically stable state in a way similar to the conditions stipulated by the thermodynamic hypothesis for protein folding, while also being recyclable for use in future configurations.

The system is capable of choosability and scalability due to its mechanical approach: the tiles can be made from other materials so long as they do not prevent magnetic interactions, and magnetic forces scale downwards well with respect to tile dimensions. Addressability, which relates to the alignment

process of tiles, will be covered in future work when we develop the process initially assembling tiles into a chain of the correct sequence before actuation. Instead, we primarily investigated system programmability by producing tiles with an identical external geometry that allows the production of a variety of different structures. Potential applications of the system outputs include usage as self-repairable meta-materials and meta-structures, or alternatively as a method for creating products that can be rapidly and efficiently recycled.

Data accessibility. Data, code and materials supporting this publication are available at https://www.dropbox.com/sh/ivh70cnneitw7oh/AABbhur3i3GvPD8aAs8SjfOna?dl=0 or data available from the Dryad Digital Repository at: https://doi.org/10.5061/dryad.4181mp7 [47].

Authors' contributions. S.M. conceived the concept; S.M., E.J.S. and A.M.T. designed the research; E.J.S. and V.B. performed the experiments; E.J.S., V.B. and B.L. carried out simulations, E.J.S. analysed the data; E.J.S., S.M. and A.M.T. wrote the paper; S.M. provided funding.

Competing interests. We declare we have no competing interests.

Funding. The research was supported by funding from the Department of Electronic Engineering at University of York.

Acknowledgements. We wish to thank Etienne Perroux from ESEO Angers for his improvements to the simulations, and Dandolo Flumini at Zhaw in Switzerland who provided inspiration with a similar design to that shown in figure 3(a) with horizontally oriented magnets.

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
