## [Reviewer comments · Royal Society Open Science]

Review History

RSOS-182128.R0 (Original submission)

Review form: Reviewer 1

Is the manuscript scientifically sound in its present form?

Yes

Are the interpretations and conclusions justified by the results?

Yes

Is the language acceptable?

Yes

Is it clear how to access all supporting data?

Yes

Do you have any ethical concerns with this paper?

No

Have you any concerns about statistical analyses in this paper?

No

Recommendation?

Accept as is

Comments to the Author(s)

The research described in this paper - on an approach to generating 2D structures from an initial 1D line of tiles through cascading magnetic interactions - is interesting, novel and scientifically sound. I believe it constitutes a solid contribution to the literature. The paper is well written, the motivations are clear and the authors show a proper understanding of the strengths and weaknesses of the approach.

In my opinion the authors have more than adequately responded to the original reviewers' comments.

As a biologically inspired approach to structure formation this is interesting and potentially important work. The authors have produced a cleverly designed experimental framework that has been used to demonstrate that the basic principles of their approach do indeed work.

Review form: Reviewer 2

Is the manuscript scientifically sound in its present form?

Yes

Are the interpretations and conclusions justified by the results?

Yes

Is the language acceptable?

Yes

Is it clear how to access all supporting data?

Yes

Do you have any ethical concerns with this paper?

No

Have you any concerns about statistical analyses in this paper?

No

Recommendation?

Accept with minor revision (please list in comments)

Comments to the Author(s)

The nomenclature used in this paper is not an easy one to follow for Biological Researchers. In fact there are statements that might seem a bit off.

Take for example

"One example of biological self-assembly can be found in the assembly of DNA nucleotide bases

into the correct A-T and C-G pairs in a highly stochastic process, demonstrating a high level of addressability."

I would argue that DNA replication is NOT a stochastic process (unless you consider any chemical reaction a highly stochastic process).

A few other examples of this type of nomenclature clash exist and should be clarified.

Nomenclature should also be defined early on if biologists are to engage with the paper from the beginning.

There are also some typos and the manuscript should be passed through a word corrector (e.g. reparable should be repairable).

Finally, I would like to see some films of experiments with more than three tiles. It is stated that you performed experiments with up to 10 tiles, so why not show some of these, with 5 or 6 tiles?

Decision letter (RSOS-182128.R0)

04-Jun-2019

Dear Professor Southern

On behalf of the Editors, I am pleased to inform you that your Manuscript RSOS-182128 entitled "Catalytic Self-Folding of 2D Structures through Cascading Magnet Reactions" has been accepted for publication in Royal Society Open Science subject to minor revision in accordance with the referee suggestions. Please find the referees' comments at the end of this email.

The reviewers and handling editors have recommended publication, but also suggest some minor revisions to your manuscript. Therefore, I invite you to respond to the comments and revise your manuscript.

- Ethics statement

- Data accessibility

<http://datadryad.org/submit?journalID=RSOS&manu=RSOS-182128>

- **Competing interests**

- **Authors' contributions**

- **Acknowledgements**

- **Funding statement**

Because the schedule for publication is very tight, it is a condition of publication that you submit the revised version of your manuscript before 13-Jun-2019. Please note that the revision deadline will expire at 00.00am on this date. If you do not think you will be able to meet this date please let me know immediately.

on behalf of R. Kerry Rowe (Subject Editor)
openscience@royalsociety.org

Editor Comments to Author:

Please accept our apologies for the unusual length of time that this transferred manuscript has taken to complete review: we struggled to secure the advice of the more critical of the original reviewers, but appreciate the support of the two reviewers who have provided commentary. Broadly, your paper appears to be ready for acceptance, though one of the reviewers provides feedback that we'd like you to address in your revised paper.

Reviewer comments to Author:

Reviewer: 1

Comments to the Author(s)

The research described in this paper - on an approach to generating 2D structures from an initial 1D line of tiles through cascading magnetic interactions - is interesting, novel and scientifically sound. I believe it constitutes a solid contribution to the literature. The paper is well written, the motivations are clear and the authors show a proper understanding of the strengths and weaknesses of the approach.

In my opinion the authors have more than adequately responded to the original reviewers' comments.

As a biologically inspired approach to structure formation this is interesting and potentially important work. The authors have produced a cleverly designed experimental framework that has been used to demonstrate that the basic principles of their approach do indeed work.

Reviewer: 2

Comments to the Author(s)

The nomenclature used in this paper is not an easy one to follow for Biological Researchers. In fact there are statements that might seem a bit off.

Take for example

"One example of biological self-assembly can be found in the assembly of DNA nucleotide bases into the correct A-T and C-G pairs in a highly stochastic process, demonstrating a high level of addressability."

I would argue that DNA replication is NOT a stochastic process (unless you consider any chemical reaction a highly stochastic process).

A few other examples of this type of nomenclature clash exist and should be clarified.

Nomenclature should also be defined early on if biologists are to engage with the paper from the beginning.

There are also some typos and the manuscript should be passed through a word corrector (e.g. reparable should be repairable).

Finally, I would like to see some films of experiments with more than three tiles. It is stated that you performed experiments with up to 10 tiles, so why not show some of these, with 5 or 6 tiles?

Author's Response to Decision Letter for (RSOS-182128.R0)

See Appendix A.

Decision letter (RSOS-182128.R1)

11-Jul-2019

Dear Professor Southern,

I am pleased to inform you that your manuscript entitled "Catalytic Self-Folding of 2D Structures through Cascading Magnet Reactions" is now accepted for publication in Royal Society Open Science.

on behalf of R. Kerry Rowe (Subject Editor)
openscience@royalsociety.org

Follow Royal Society Publishing on Twitter: [@RSocPublishing](https://twitter.com/RSocPublishing)
Follow Royal Society Publishing on Facebook:
<https://www.facebook.com/RoyalSocietyPublishing.FanPage/>
Read Royal Society Publishing's blog: <https://blogs.royalsociety.org/publishing/>

Appendix A

Response to Decision Letter

Thank you both for your helpful comments, we are very grateful for your insight.

Response to Reviewer 1

It appears that this reviewer is satisfied with the updates we have made to the work since our first submission. We are thankful for their guidance in this respect.

Response to Reviewer 2

There was some confusion with regards to the attached manuscript containing corrections from Reviewer 2. After discussion with Alice Power it has been confirmed that the comments made in the manuscript should be ignored and any further corrections should be made based on the suggestions from the main body of the decision letter. The following updates have been made:

- *Describing the DNA replication as a stochastic process* - we have removed the word 'stochastic' in this context as we admit it is misleading. We have changed the sentence

“One example of biological self-assembly can be found in the assembly of DNA nucleotide bases into the correct A-T and C-G pairs in a highly stochastic process. “

to

“One example of biological self-assembly can be found in the assembly of DNA nucleotide bases into the correct A-T and C-G pairs in a thermally dynamic process involving massive sampling of arrangements”.

- We also recognise that Section 1.5 Anfinsen's Thermodynamic Hypothesis is based on a classical theory and there have since been several improvements and counter-examples to it from biological research. We have made the following updates to our text to reflect this:

“Anfinsen showed that, with some exceptions, all of the information required for a protein amino acid sequence to fold is contained with the chain itself and that the globally stable (native) state of a sequence must be located at the sequence's energetic minimum.”

has been changed to

“Anfinsen was the first to hypothesise that all of the information required for a protein amino acid sequence to fold was contained with the chain itself and that the globally stable (native) state of a sequence must be located at the sequence's energetic minimum. ”

We have also added the following sentence after the list of system requirements:

“While Anfinsen's hypothesis is now widely considered to be a simplification of biological processes and that several counter-examples to his proposed system requirements exist, we believe that his statements are a useful starting point for our approach in micro-robotics.”

- *There are also some typos and the manuscript should be passed through a word corrector (e.g. reparable should be repairable)* - This has been carried out.
- *Finally, I would like to see some films of experiments with more than three tiles. It is stated that you performed experiments with up to 10 tiles, so why not show some of these, with 5 or 6 tiles?*
 - There is a video of an experiment containing 7 tiles with the latest submission.

Again, the authors are very grateful for the time and thought that both reviewers have put into assessing our work and making it worthy of publication.